# 3D Printing of Habitats on Mars: Effects of Low Temperature and Pressure

**DOI:** 10.3390/ma16145175

**Published:** 2023-07-23

**Authors:** Reza Hedayati, Victoria Stulova

**Affiliations:** Department of Aerospace Structures and Materials (ASM), Faculty of Aerospace Engineering, Delft University of Technology (TU Delft), Kluyverweg 1, 2629 HS Delft, The Netherlands; v.v.stulova@student.tudelft.nl

**Keywords:** Mars, additive manufacturing, 3D printing, regolith, mechanical properties

## Abstract

Due to payload weight limitations and human vulnerability to harsh space conditions, it is preferable that the potential landing location for humans has an already constructed habitat preferably made from in situ materials. Therefore, the prospect of utilizing a readily available Martian material, such as regolith, in an easily programmable manufacturing method, such as 3D printing, is very lucrative. The goal of this research is to explore a mixture containing Martian regolith for the purposes of 3D printing in unfavorable conditions. A binder consisting of water and sodium silicate is used. Martian conditions are less favorable for the curing of such a mixture because of low temperature and pressure on the surface of the planet. In order to evaluate mechanical properties of the mixture, molding and 3D printing were conducted at various curing conditions and the mechanical and physical characteristics were compared. Due to the combination of low reaction speed at low temperature (2 °C) and rapid water evaporation at low pressure (0.1–0.01 bar), curing of the specimens in Martian conditions yielded unsatisfactory results. The reaction medium (water) evaporated before the curing reaction could progress enough to form a proper geopolymer. The specimens cured at high temperatures (60 °C) showed satisfactory results, with flexural strength up to 9 MPa when cured at a temperature of 60 °C and pressure of 1 bar. The specimens manufactured by 3D printing showed ultimate flexural strength that was 20% lower than that of equivalent molded specimens. Exploring potential mixture modifications and performing improved tests using the basis laid in this research can lead to an effective and realistic way of utilizing Martian regolith for unmanned 3D-printing purposes with minimal investment.

## 1. Introduction

Since the Apollo 17 mission back in 1972, only highly specialized robots have been sent to other celestial bodies for in situ exploration, as it is considerably safer and requires little supportive payload. Robots, however, are quite limited in their functionality, as they are used for a particular number of scientific goals, programmed during the robot development [1]. Sending humans instead of robots would provide significantly more flexibility in research and valuable first-hand experience in the location to be explored. However, human lives are considerably more valuable than robots in case of a catastrophic failure, and they require a lot of equipment in the form of food, water, and shelter, with all living necessities and space radiation protection [2,3]. Due to the latter, in particular, the time a human can spend in a space environment is very limited [4]. All these basic needs already force the mission to be considerably more expensive as compared to an unmanned one. Hence, to make the most of a manned mission, as much preparation as possible should be done before sending a human to another celestial body [5].

Because of the large amount of sustaining equipment required for a manned mission, together with the necessary scientific equipment, the expected size of the payload to be taken for a manned mission is very high [6]. Therefore, as many in situ resources as possible have to be utilized for mission cost reduction as well as maintenance convenience [7]. In addition, preparing the landing site for astronauts before they arrive can significantly increase the amount of useful research work that can be undertaken in a set period of time, as space radiation limits the amount of time that humans can spend on other celestial bodies. Therefore, 3D printing can be looked at as a promising manufacturing method for such a purpose, as the desired design can be programmed in the printer and simply executed, provided that printing material is supplied [8,9,10,11,12,13,14,15]. If this printing material contains mostly in situ gathered materials, the amount of necessary payload can be significantly reduced [16].

This research concerns investigating the printing mixtures that can be utilized for Martian 3D-printing purposes, and selecting the most promising one. This mixture is then tested for overall viability using different manufacturing methods, namely molding and printing, and the conclusion regarding the overall idea is made. As the starting point, Martian regolith simulant is taken as the primary material, as the actual Martian regolith is easy to collect and process from the surface of the planet. The main goal of the research is to find and develop a suitable binder/regolith mixture for Martian regolith 3D-printing application and, after a proper investigation, reach a conclusion about the viability of the said mixture. In order to achieve this goal, various specimens were constructed in different conditions, and the properties of the specimens were compared in terms of physical and mechanical properties.

## 2. Differences between 3D Printing on the Earth and Mars

Transporting any payload into space is very expensive as compared to transportation costs on Earth. While there is a wide range of materials such as polymers [17,18,19,20,21], metals [22], shape memory polymers [23,24], bio-inks [25], and elastomers [26] that can be used for 3D printing on Earth, only a very limited quantity of them can be reasonably transported into space. Transporting a large amount of building material to build one or multiple habitats on Mars seems unrealistic, especially considering the fact that the walls of such a habitat need to be thick and strong to protect humans from dangerous environmental conditions, such as meteoroids and space radiation, from which the Martian atmosphere offers negligible protection. Hence, it is necessary to use in situ resources when constructing the habitat and potentially utilize in situ resources for synthesis of usable binders. By exploring the underground resources of the planets, it could be possible to dig and construct an underground shelter, which offers high protection from meteoroid strikes and radiation due to the potentially high wall/ceiling thickness. However, for this to become possible, more exploration is required. If the raw resources are not usable, the possibility of synthesizing a printing material would be open once more information about the availability of those resources is better known. This would require extra energy and possibly complex equipment, but the result might make the Martian colony independent from Earth.

As mentioned above, due to the near-absence of atmosphere on Mars, small space objects such as meteors and asteroids cannot be incinerated by passing through the atmosphere, as happens around Earth. On Mars, 30,800 kg of meteoroid mass strikes the surface per day [27]. Earth receives considerably less meteoroid mass (between 2900 and 7300 kg per year, excluding dust particles), which is also slowed down considerably by the atmosphere [28]. The danger of meteoroid impact persists not only for potential humans and habitats on Mars, but also for the 3D-printing equipment and unfinished habitat. The printing equipment itself can contain vulnerable components they might be damaged by smaller strikes. The initial stages of printing, which are expected to be most vulnerable due to unfinished sheltering structures and hence exposed printing equipment, are expected to be performed without supervising humans, eliminating the possibility of easy repair. Material used for building the habitat should also have enough impact resistance to withstand small strikes.

Unlike on the Earth, where the presence of the atmosphere and water stabilizes the temperature in each region, Mars has large temperature fluctuations during the day/night cycles. Martian surface experiences a swing from −153° to +20° near the equator. Near the poles, just like on Earth, day/night cycles are considerably longer; hence, the temperature swings are smaller. Temperature affects 3D printing and material curing to a large extent, and therefore printing machines, processes, materials, and habitat location have to be established together, as they affect each other closely. Near-vacuum conditions due to the thin atmosphere make the problem even worse, especially if the binder material used to solidify Martian regolith is a liquid. On Earth, the evaporation behavior of liquids is governed by the presence of water vapor in the atmosphere. However, atmosphere is nearly absent on Mars, and hence no water vapor is present. This would result in deposited liquid rapidly evaporating into the ambient surroundings and, immediately afterwards, freezing due to low temperatures. A summary of the challenges of 3D printing on Mars is presented in Table 1.

## 3. Materials and Methods

### 3.1. Regolith Characterization

The density of regolith was measured inside a measurement cylinder filled with water. The initial water volume was recorded inside the measuring cylinder, which was placed upon the scales. A portion of regolith was then added to water, and the weight difference together with the volume difference was recorded. From these measurements, density was calculated by $\rho =\frac{m_{F}-m_{I}}{V_{F}-V_{I}}$, where $m_{F}$ and $m_{I}$ respectively stand for the final and initial masses of the mixture, and $V_{F}$ and $V_{I}$ stand for the final and initial volumes of the mixture, respectively. The average density of wet regolith was 1873 ± 128.4 kg/m^3^. This is comparable to the density of wet sand which equals 1922 kg/m^3^. The average density of wet regolith is used as the reference density for comparing the specimen densities and calculating relative densities, which in turn will contribute to analysis of mechanical properties.

Particle size and shape can affect the reaction parameters such as dissolution within the binder, as well as restrictions on temperature and pressure. Small and smooth particles are expected to spread within the binder mass at a higher rate with lower effort, and hence decrease the minimum amount of binder required for paste formation. At the same time, in the conditions where the pressure is low and ambience contains no water vapor, the binder solution evaporation happens at a considerably higher rate. Low temperatures cause preliminary freezing of the binder and halt the curing reaction. Large and rough particles might help trap the binder within the paste and prevent it from escaping into the ambient surroundings and subsequently freezing, thus expanding the temperature and pressure reaction window. Considering that conditions on Mars involve low temperatures and pressures, it is important that these considerations are taken into account.

The regolith particle dimensions were analyzed using a dry sieve test. A bulk of regolith was sieved through 125 µm and 63 µm sieves, and the resulting weight percentages were measured. Overall, it can be seen in Table 2 that the majority of particles are smaller than 125 µm.

To investigate the particle shapes, scanning electron microscopy (SEM) imaging was used. Together with SEM, energy-dispersive X-ray spectroscopy (EDX) analysis was also performed. Sieved powder specimens were placed upon a layer of carbon paint in order to make the imaging possible and also to fix the positioning of powder particles over the base cylinder, so that contamination of the chamber could be avoided.

From the SEM images (Figure 1), it can be seen that, in general, particles have irregular shapes, ranging from nearly spherical to highly elongated. The larger the particles, the higher the chance that the shape deviates from a sphere. Particles of medium and small size are closely packed, while large particles are mostly located apart from each other (Figure 1). Among the medium and small particles, porosity can be observed, meaning that liquid can potentially get between the particles easily without considerable mixing required, and initiate the curing reaction.

EDX measurements were also performed during SEM measurements, and the results are presented in Figure 2. The analysis showed a high percentage of carbon and some percentage of gold, which was the result of carbon paint and gold coating, which are necessary for SEM measurements. Therefore, an EDX result adjustment was performed (listed in the table of Figure 2) and carbon and gold were excluded from the calculations. Copper and zinc were also excluded from calculations, as the presence of these is a result of carbon paint being too thin or cracked in some locations. As it can be seen from the composition, a large amount of silicate oxide and aluminum oxide is present in regolith, which indicates its suitability for a geopolymer base, meaning that geopolymer binder can be investigated among other options.

As expected, oxygen is indicated as the most present element, which can be attributed from the fact that regolith mostly consists of various oxides. Among the trace elements, potassium and titanium are detected by EDX, and it is possible that copper, zinc, carbon, and gold are also among the trace elements. Overall, the EDX analysis confirms the composition as indicated by the manufacturer.

### 3.2. Binder Selection and Characterization

Manned Mars exploration has been a research subject for many decades. Among other aspects, potential building materials have been looked at by various articles, mostly very recently. Upon the review of these articles, several main options were chosen for the binder trade-off process. These options were molten sulfur [29], Portland cement [30], geopolymer cement [31], and polymers such as polyimides and epoxies [32]. After taking into account several factors such as processing, feasibility for incorporating into 3D printing, state of research, robustness, transportation cost, and total cost, geopolymer cement (sodium silicate, ${Na}_{2}{SiO}_{3}$) was selected as the binder material.

#### 3.2.1. Theory

In granular form, sodium silicate (${Na}_{2}{SiO}_{3}$) is a white solid powder with particle size of approximately 0.5–1 mm. The powder has a melting point of 1088 °C and a density of 2.61 g/cm^3^ at 20 °C. This powder is soluble in water, but the solubility highly depends upon the ratio of powder to water and the water temperature. The solubility in this context is meant to be complete when the solution is fully transparent, and no undissolved powder particles are visible. Between 25 °C and 80 °C, solubility varies between 22.2 g per 100 mL (18% concentration) and 160.6 g per 100 mL (61.6% concentration). Time is also an important variable when making the binder solutions, because the duration required for dissolution increases as temperature decreases and the desired concentration increases. Stirring at 360 rpm and 80 °C temperature was applied to increase the solution speed in the binder material. 

The geopolymer reaction consists of five major steps, which are schematically demonstrated in Figure 3 [33]. For this reaction to be initiated, an aqueous alkaline medium (hydroxide or silicate) has to be combined with a solid component that includes aluminates and silicates, preferably in powder form. Upon mixing these two components, in this case regolith powder and liquid binder solution, the reaction initiates. 

Dissolution is a stage where aluminosilicate particles from regolith powder start dissolving in alkaline medium, provided by sodium silicate solution. As the result of this phase, monomers of aluminates and silicates are formed. This reaction step consumes water to produce the reactive particles. General formulae of these hydrolysis reactions for aluminates and silicates, where only mass and charge balances are taken into account, are shown as follows:(1)$$Al_{2}O_{3}+3H_{2}O+2OH^{-}\to 2Al{\left(OH\right)}_{4}^{-}$$
(2)$$SiO_{2}+2OH^{\_}\to {\left[SiO_{2}{\left(OH\right)}_{2}\right]}^{2-}$$
(3)$$SiO_{2}+H_{2}O+OH^{-}\to {\left[SiO{\left(OH\right)}_{3}\right]}^{-}$$

Speciation equilibrium, also described as solution saturation in this case, is reached faster with higher pH values. As the solution becomes saturated with the particles from the previous step and subsequently oversaturated, aluminosilicate particles combine with each other to nucleate and form small oligomers. During this process, water condenses. It can either evaporate if the reaction occurs close to the surface, or it remains trapped within the pores of material bulk. 

Gelation happens as more and more monomers combine together into oligomers, and hence the mixture becomes less and less liquid. This forms what is referred to as the Gel 1 phase, which consists of a large number of small oligomers. These oligomers continue to react with each other, releasing water into the system, but eventually, because of restricted mobility, no nearby connections can be established. At this point, the formed oligomers begin to reorganize in order to form more connections within the polymeric network. At the same time, as aluminum particles dissolve considerably faster than silicon ones, the silicon groups continue to dissolve and participate in the polymerization process. This process continues to harden the material bulk, forming a more stable shape in the Gel 2 phase. The last stage of the polymerization reaction (with potassium instead of sodium, which also can be used for this purpose) is shown in Figure 4. The speed of this phase depends upon ambient conditions and initial component compositions. For example, unfavorable balance between aluminum particles, silicon particles, alkaline medium, and water might lead to gel not even forming in the first place.

It should be noted that the various above-mentioned reaction steps occur concurrently in different locations of the bulk, and do not progress strictly in order simultaneously in the whole bulk [35]. For example, initial gel precipitation starts when the solution in a particular location reaches saturation. However, other locations can be still reorganized to achieve this. This can be viewed from the microscopic image in Figure 5, where, in some locations, the material still flows to reach the destination (mostly on the right), while other locations have reached saturation and have already started the gelation phase (mostly on the left). Hence, it is important to provide sufficient time for the mixture to complete the reaction process before using the product, as the overall reaction progression can vary in different bulk locations. Furthermore, varying the reaction progression between the regions can introduce inhomogeneity in the bulk, as less links between the regions can be formed in the end.

#### 3.2.2. Measurements

To investigate the curing reaction properties, differential scanning calorimetry (DSC) analysis was conducted by curing the specimens with the same composition and various temperatures. These temperatures are those at which actual specimens would be cured. This DSC analysis allowed for a better estimation of the times in which initial gelation and final hardening take place, which contributed to the optimal specimen processing. Removing the specimen after the shape is established, but before complete drying and hardening, allows for easy removal of the specimen from the mold without the risk of damaging it.

DSC experiments were conducted for the same temperatures that were selected for curing (2 °C, 23 °C, and 60 °C). It was expected that low-temperature curing would take a long time; hence, the measurement was conducted overnight [36]. The DSC method used for this measurement consists of decreasing the temperature from room conditions to 2 °C at the rate of 5 °C/min (default rate), holding the specimen at 2 °C for 12 h, and heating it back up to room temperature at the rate of 5 °C/min. The second DSC test (at 23 °C) was conducted simply by holding the specimen at room temperature of 23 °C for 4 h. Knowing that high-temperature curing occurs considerably faster, the selected method was to heat up the specimen from room temperature of 23 °C to 60 °C at a standard rate of 5 °C/min, holding it for 2 h, and dropping the temperature back to room conditions. 

### 3.3. Molding Process

Although the research was primarily concerned with 3D printing the regolith paste, molded specimens serve as a reference point in the evaluation of the results. Through molding, it is more convenient to see the dependency of mechanical properties of cured specimens in ambient conditions due to the lower number of processing parameters. In addition, it is easier to change the paste compositions when it comes to molding, as varying the thickness of the paste has a very high impact on printing process.

Brick-shaped molds with specimen thicknesses of t=5 mm, width of w=10 mm, and length of l=30 mm were 3D printed using PLA (polylactic acid) material. Specimens cured at low temperature were stored in a refrigerator, specimens for curing at room temperature and pressure were stored in the laboratory, while specimens for curing at low pressure and high temperature were stored in an oven with the corresponding temperature set beforehand. Low pressure was reached after placing the specimens in the oven. During the second curing stage, while the specimens were still soft but had a set shape, they were removed from the molds.

For the purpose of finding the optimal binder/regolith combination for best mechanical and physical performance, multiple parameters had to be considered to generate a better overview of the properties. In the end, four parameters were selected: temperature, pressure, binder concentration, and regolith-to-binder ratio.

First of all, molding and curing specimens at room condition is the simplest case to implement, as no additional equipment is required. Therefore, basic specimens were also printed in room conditions. Because of this, room-condition-cured specimens served as a baseline for mixture property evaluation, and a higher number of specimens were manufactured for room condition tests as compared to other cases. Regarding the binder and mixture concentrations, there are restrictions that define the boundary conditions. When it comes to regolith concentration, at 70%, the mixture was hardly mixable, and it took significant effort to cover all the powder particles with liquid. At 60% regolith concentration, however, the mixture formed very easily within a couple of stirs, and all the powder was covered in liquid. During the preliminary testing, an even lower concentration of 55% regolith was tried, but a large number of cracks were observed after the curing process, and the resulting overall mechanical robustness was very low. In addition, in real applications, it is desired to have a regolith concentration that is as high as possible for reduced transportation costs. Hence, 60% concentration was deemed the lowest to be investigated (see Table 3).

The binder concentration boundaries were determined through binder solubility information and preliminary testing. The solution with concentration of 60% was not completely formed after 30 min of stirring at 80 °C, and increasing the temperature led to quick liquid evaporation, skewing the resulting concentration. A solution having a concentration of 55% was able to be formed, but it did not last in completely transparent form for longer than 3 min. This was the highest solution concentration boundary. At 30% concentration, the preliminary testing specimens were mechanically extremely weak after curing; hence, this was selected as the lower bound for the tests. After the mechanical testing of specimens cured at 60 °C, the binder concentration of 30% was eliminated because of insufficient strength compared to higher concentrations. Binder having a concentration of 55% was also eliminated because of fast solution crystallization as well as reduced mechanical properties as compared to the 50% solution. As fast crystallization produces unreliable results, it was decided to skip this binder concentration in further experiments, see Table 4.

### 3.4. 3D Printing Process

Molded specimens provided a large variety of data regarding mixture composition properties and their influence at ambient conditions. While some conclusions about the mixture viability can be made from the molding process, it is useful to see how such a mixture can perform at similar conditions, but in the 3D-printing process. The printer used for this project was an Automated Dispensing Robot TSR2301 (Techcon System, Cypress, CA, USA) (Figure 6a). Primarily used for dispensing fluids, as it can operate within three axes, and it was used as a binder deposition 3D printer after making some adjustments.

First of all, the viscosity of the paste is of vital importance when it comes to printing, especially with such a limited outlet diameter. A low-viscosity mixture (65% regolith and below) did not hold the shape upon deposition at all, flowing all over the build plate. Mixture with the lowest possible regolith ratio that could hold the programmed shape was the one with 66% regolith concentration, which set the lower boundary for printing. Because of rapid viscosity change with respect to regolith ratio, going up to 69% regolith content resulted in a paste that was too thick to be printed even with the largest syringe diameter possible (7 mm; Figure 6b). 

The syringe provided by the manufacturer (with maximum diameter of 3 mm) was redesigned in order to achieve two goals: increase the syringe outlet diameter, and increase the slope inside the syringe, as the horizontal bottom wall of the originally provided syringe causes the material to be pressed against it and cure under pressure instead of being pushed into the outlet. The issue of rapid mixture curing under pressure significantly complicated the printing process. Although the syringe redesign helped to partially alleviate the issue, the problem was not completely fixed. The issue was alleviated by significantly increasing the outlet diameter and decreasing the mixture viscosity, but the latter option is limited because its leads to inability of the deposited paste to keep its shape upon deposition. The best combination of deposition conditions that could be achieved was deposition at 0.3 bar pressure provided by the fluid dispenser, with the outlet diameter of 7 mm.

The specimen printing pattern consisted of four lines: two next to each other with two more placed on top (Figure 6c). Two separate sets of printing programs were created. The first was printing all the four lines in one go, while in the second program, a delay was allowed between the first and second layer of deposition. This variation can provide information about the impact of printing duration on the mechanical properties of the resulting part. Theoretically, having layers deposited with a delay causes them to have different progressions in their curing process, which can introduce inhomogeneity into the material bulk through not allowing polymeric links between the layers to form. The potential decrease in the mechanical properties of the specimens printed with a delay can potentially lead to the conclusion that printing has to be performed as fast as possible or with specific designed printing paths to ensure good properties. When it comes to the printing projects having a large scope, namely habitats, the inevitable printing delays might create issues and hence have to be taken into account.

### 3.5. Mechanical Tests

The elastic modulus is defined as the ratio between the applied stress and the specimen strain. In the case of the three-point bending test, the elastic modulus can be determined by taking a linear portion of the force–displacement curve in the three-point bending test (Figure 7), and then dividing the force values into stress using the three-point bending formula for stress (ASTM C78 Standard [38]):(4)$$\sigma =\frac{3Fl_{s}}{2wt^{2}}$$
where F is the applied force value, $l_{s}$ is the testing bench span, w is the specimen width, and t is the specimen thickness. The displacement d was converted to strain values using equation $\varepsilon =\frac{d}{t}$.

## 4. Results 

### 4.1. DSC Results

From the obtained plots for 2 °C (Figure 8a,b), the reaction seemed to be completed within 100 min of the start of the experiment, which can be explained by the very small specimen size (~11 mg). After 100 min, the curve becomes almost horizontal. Figure 8b covers the part of the result plot that contains the reaction space. As it can be seen, the second part of curing (hardening) lasted for 92 min. The first part of curing happened considerably faster, as it can be seen in Figure 8a, which covers the very beginning of the experiment. The increase in heat flow happened within 2 min, which is quite fast.

The result of the 23 °C case, from the start until the point of curve straightening, is shown in Figure 8c. Unlike in the case with a low temperature, the first gelation phase starts almost immediately upon loading the sample in the DSC apparatus, as the beginning of the drop is at 1 min. Seemingly, the first phase here lasts longer (19 min) than in the case with low-temperature curing. After that, the second phase lasts for approximately 1 h. It is important to note that the specimen size in this case was approximately 21 mg, which could explain the duration of the first phase. As it can be seen in Figure 8d, the reaction of 60 °C was completed under 20 min. The first gelation step ended within 5 min, while the second gelation phase lasted for only 13 min. 

The results show that the second gelation phase, or hardening of the mixture, occurs considerably faster at higher temperatures. Indeed, in this phase, oligomers rearrange themselves in order to find more available connections and form a complete 3D network, where mobility plays a crucial role. For the first phase, however, this dependency does not seem to be the case. Looking at the beginning of DSC results for low-temperature curing (Figure 8a,b), it can be noticed that the curve starts with a straight line, while the other two plots start with a sharp rise. This can be explained as follows: as the curing for the 2 °C specimen started at room temperature and then the temperature dropped, the heat flow change from the temperature drop cancelled out the part of the curve related to the first stage of the curing. Regardless, looking at the reaction, it is clear that the first gelation stage, where small oligomers form, depends more on the concentrations of necessary components and alkalinity of the mixture, and temperature is not as important due to the assumed homogeneous distribution of the components. The reaction there can occur straight away, without much movement being necessary.

### 4.2. Molded Specimens

#### 4.2.1. Physical Properties

A stark contrast between different specimens already arose during their removal from molds. In fact, one of the reasons to remove specimens during the curing process and not afterwards was the danger of destroying the specimens. This was especially noticeable for the specimens cured in the oven at a high temperature (60 °C). The specimens also expanded at temperatures higher than they did during deposition, which was done at room temperature (23 °C). As a result, the specimens were not removable from the mold without forcefully separating them from the walls with a thin object such as a razor. The specimens were removed from the mold during the second stage of curing, the timing of which was derived during DSC analysis. During the first reaction stage, the mixture was not yet gelated, meaning it lost shape upon the attempt to remove it.

When it comes to curing time, temperature is the detrimental factor, while pressure seems to have little to no effect on it. For specimens cured at 60 °C, it was easy to see that the total curing time was around 6 h for all three different pressure conditions. The state of specimen curing after 6 h was identical by the end of this time, and the progress was checked every 30 min. In comparison, the total curing time for room temperature specimens was around 30 h. The total curing time for specimens at the lowest temperature (2 °C) was 72 h, which is considerably longer.

Porosity of the specimens was directly related to curing pressure of the specimens, as it can be seen in Figure 9. Curing completed at atmospheric pressure (Figure 9a–c) did not induce any visible porosity in the specimens. Specimens cured at 0.1 bar (Figure 9d,e) had small uniform pores throughout all the specimen. Specimens cured at even lower pressure of 0.01 bar (Figure 9f,g) had several large pores instead of a large number of small ones. Additionally, the surfaces of these specimens were rugged and irregular. The significant increase in the visible porosity of the specimens cured at decreased pressure can be explained by the rapid decrease in the boiling temperature of the mixture as the pressure is decreased. The decreased content of moisture in the ambient surroundings resulting from the vacuuming process causes the liquid present in the mixture to rapidly evaporate. This can explain the difference between pores in the specimens: slightly lowered pressure (0.1 bar) makes boiling slower and less rapid, generating small and uniform bubbles. Very low pressure (0.01 bar, slightly higher than Martian pressure), on the other hand, causes the boiling process to be very rapid, making bubbles large and also making the surface of the specimens rugged. 

Low pressure (0.01 bar and 0.1 bar) combined with low temperature (2 °C) led to specimens that could not keep their shape due to poor mechanical properties and robustness. That is why the physical and mechanical results of the specimens manufactured at a temperature of $T_{c}=2℃$ and under pressures of $P_{c}=0.1bar$ and $P_{c}=0.01bar$ are not reported in this paper.

The change in the relative density of molded specimens with respect to the change in $Na_{2}SiO_{3}$ concentration, regolith percentile, pressure, and temperature can be seen in Figure 10. Increasing the $Na_{2}SiO_{3}$ concentration in ambient pressure and P = 0.1 bar increased the relative density, while it had a negative effective at P = 0.01 bar (Figure 10). In general, increasing the temperature decreased the relative density of the final product (Figure 10). Higher temperatures force evaporation to occur faster and improve overall molecule mobility; so, during the curing process, when mobility of the oligomers is still present, more residual water molecules are able to escape the material bulk and evaporate.

To have better information about the pressure effect on density, tests at more conditions would be useful. At high temperature, clearly, lowering the pressure decreases the density as water present in the mixture was forced to evaporate into the near-vacuum ambient surroundings (Figure 10). This creates pores, which become permanent in the bulk of the material because of fast curing times. These pores are visible with the naked eye in Figure 9g. However, as shown in Figure 10, such dependency may not be the case for other temperatures. While specimens at a pressure of 0.01 bar still show the lowest density, specimens cured at 0.1 bar clearly have the highest average relative density (Figure 10). Potentially, this could be explained by the balance of the reaction rate and the vacuum-induced boiling rate. The specimens cured at a pressure of 0.1 bar at room temperature have very small pores (around 0.1–0.5 mm), while specimens cured at a pressure of 0.01 bar have large ones (1 mm). Perhaps when the reaction rate is slow enough (in lower temperatures), there is enough time for the curing oligomers to move and fill the space freed up by the boiling process. This reduces the water amount in the specimens and also reduces the number of pores. At lower pressure (P = 0.01 bar), however, the pores are simply too large to be easily filled by the curing material, as the mobility of oligomers is not unlimited. To further test this claim, however, more tests have to be conducted on a wider range of temperatures and pressures.

On average, it seems that lowering the amount of regolith decreases the density of the specimens. This can be easily explained by the increased water content, as less regolith means a greater amount of binder solution in the mixture. A higher amount of water overall means that more specimen mass can be lost to the ambient surroundings in the form of water, thereby decreasing the density. The same reasoning can be applied to the sodium silicate concentration within the binder solution, where lower concentration means higher water content that is free to be evaporated.

#### 4.2.2. Elastic Modulus

The elastic modulus values of specimens molded in different conditions are compared in Figure 11. It should be pointed out that some of the plots have a small number of data points. As indicated in Figure 9, specimens cured at certain conditions were extremely fragile and broke either during mold removal or upon the pre-loading stage (0.5 N) of the three-point bending test; hence, those specimens can be considered as failed specimens. 

As it can be seen in Figure 11, increasing the temperature clearly causes the elastic modulus to rise. The difference in elastic modulus for pressure of 1 bar between low and room temperature is not large (∼20 MPa), while the difference between room temperature and high temperature for all pressures is very large, ranging from doubling for pressure of 0.1 bar to quadrupling for pressure of 1 bar. High temperature increases the mobility of the oligomers, allowing them to interlink into polymers more reliably. This causes the stiffness to rise, as an interlinked polymer is considerably less mobile than a collection of oligomers that can slide along each other.

When it comes to pressure, elastic modulus decreases with the drop in pressure (Figure 11). For instance, at 60 °C, the average stiffness of the specimens manufactured at 1 bar is almost twice that of those made at 0.1 bar. Similar to the reasoning mentioned for the temperature effect, the presence of water during the curing allows for better interlinking into large polymeric chains, increasing the elastic modulus. As the pressure is decreased, water evaporates and restricts mobility of oligomers during the curing process, halting polymerization. Between 0.1 bar and 0.01 bar, nearly all the water evaporates quite fast, with the only difference being the size of pores. This might explain the similar elastic modulus values for both pressures, as the duration of window for polymerization is reduced similarly. As a smaller quantity of polymeric chains allows oligomers to move along each other during stress application, the elastic modulus drops.

#### 4.2.3. Ultimate Flexural Strength

The ultimate flexural strength (UFS) values for each of the molded specimens are plotted in Figure 12. Clearly, increasing the curing temperature directly increases the maximum flexural strength endurable by the part. High temperature allows for higher oligomer mobility and reaction rate, which increases the quantity of polymeric links and hence makes the material stronger. The increase in UFS values between temperature steps at each pressure ranged from 83% (such as between 23 °C and 60 °C at 1 bar pressure), to 425% (such as between 23 °C and 60 °C at 0.01 bar pressure). This is in accordance with what was observed for the elastic modulus (Figure 10). Faster water evaporation is also an effect of high temperature. Water emitted during the curing reaction in a high-temperature environment is removed from the pores through faster evaporation; however, vacant space can be filled by the material assuming it is in the process of reacting and is mobile enough, which a high temperature allows. This decreases the volume of non-functional space in the material bulk (pores become filled), and this increases the overall strength.

A decrease in the pressure induces rapid water evaporation. This generates pores, which are larger for lower pressure values, and destroys the reaction medium for link formation. The formation of large pores decreases the effective volume of the material bulk, as pores and unreacted particles do not contribute significantly when it comes to mechanical properties. The decrease in the elastic modulus also supports this observation.

### 4.3. 3D Printed Specimens

#### 4.3.1. Physical Properties

During the curing process of the printed specimens, the deposited mixture was flowing over the build plate as no walls were present to restrict the movement. This caused irregularities in the dimensions, as it can be seen in Figure 13. The printing process with a small outlet diameter creates an issue related to the pressure required for mixture deposition. Applied pressure causes the part of the mixture that does not get deposited to rapidly cure, solidifying it in the process and making it unsuitable for further printing. Roughly the same amount of mixture was prepared for each specimen print (10–11 g), but the mixture amount that was actually deposited for each specimen varied. This effect can be observed by comparing Figure 13d and Figure 13e, where in the specimen shown in Figure 13d, less mixture was deposited in the second layer, while in the specimen shown in Figure 13e, the first and second layers were both deposited similarly. By using a large syringe outlet diameter, the paste can be deposited under smaller pressure levels.

As for the differences between printing the whole specimen in one go and printing it with a delay, the difference is clear even visually. Specimens printed without a delay (Figure 13a–c) were spread over the deposition plate to a larger extent, and they were thinner as compared to their counterparts (Figure 13d–f). Specimens printed with a 30 min delay had a clear border between the layers. This indicates that the curing reaction that partially completed in the first deposited layer progresses far enough that the second deposited layer does not get fully incorporated, and mechanical properties at the border between the layers are likely different than those inside the layers. If this phenomenon affects overall mechanical properties negatively, which is likely because of the introduced inhomogeneity, it provides a stronger argument in favor of increasing the syringe outlet diameter when designing 3D-printing equipment.

The relative density, elastic modulus, and UFS of the printed samples at room temperature are compared to average values of molded specimens in each temperature and pressure condition in Figure 14. Printed specimens clearly have higher density values as compared to molded specimens (Figure 14). During the printing, the mixture was free to flow on the build plate where it was deposited. Inside the mold, there was less space available for mixture mobility. Hence, after the printing, the mixture had more freedom to rearrange into a more compact and efficient packing, considerably increasing packing density. This mobility was combined with higher surface area for water evaporation. Because of the relatively low curing temperature (room conditions), density increased as water was evaporating, thereby allowing oligomers to fill the space. With delayed printing of the second layer, there was even more opportunity for water to evaporate, as essentially some extra water had evaporated from the first layer before the second deposition, making density even higher.

#### 4.3.2. Elastic Modulus

Looking at the average values of the results of the molded specimens and the 3D-printed specimens in Figure 14b, the elastic modulus of specimens printed at room conditions is much lower than that of their molded counterparts, and it is close to the elastic modulus of specimens molded at low pressure (58 MPa for delayed specimens and 65 MPa for non-delayed, on average, as compared to 134 MPa for molded specimens). During molding, the mobility of the mixture is very limited, and deposition is performed very fast by pouring the material. Printing introduces a delay while the syringe is working (the programmed printing duration was 8 s), and the mixture spreads over the deposition location afterwards. This induces the variations in reaction progression across different bulk locations. While parts of the mixture are flowing and rearranging to achieve optimal packing, some other parts are already settled, and reaction is progressed further. Between the particles from locations with different reaction progress, fewer oligomers can interlink, hence reducing the quantity of polymeric links and decreasing the elastic modulus. The stiffness of specimens with delayed layer deposition is higher than that of those printed in one go, as more water manages to evaporate from the specimen bulk. This restricts the mobility of oligomers during stress application.

#### 4.3.3. Flexural Strength

The flexural strength trend (Figure 14c) is similar to the elastic modulus trend (Figure 14b). At room conditions, the average highest achieved flexural stress of printed specimens is 70% of the value for molded specimens. The lack of boundaries around the deposited mixture allows the paste to flow in different directions, changing the reaction progression extent across the material bulk. This reduces the quantity of polymeric links formed, as the curing material is not as homogeneous as the molded specimen. The delay between printing the layers makes the bulk even less homogeneous, decreasing flexural strength.

## 5. Discussions

### 5.1. Viability of the Products

Looking at the results as a whole, the proposed binder/regolith geopolymer mixture looks decent, especially considering the flexibility of the binder composition, even though it was outside the scope of this research. A large issue arises when considering the fact that, for 3D printing to be applicable on other celestial bodies, the ability to cure well in low pressure and temperature conditions is crucial. From this point of view, the mixture is not suitable for space application, in its current form. An overview of the explored curing conditions and their effect on the resulting material properties is provided in Table 5.

The role of water in the curing process of geopolymer mixture is crucial. It is necessary for the hydrolysis of powder particles, and then it is emitted during the polymerization process and provides the medium for polymerization to occur. Decreased pressure causes water to rapidly evaporate and leave pores in the material bulk, which ends the curing reaction prematurely and introduces inconsistency in the material bulk. Increasing the temperature can help bypass the rapid loss of water by speeding up the reaction. However, a combination of low temperature and pressure yields materials that are hardly useful for construction purposes. In this project, performing experiments with specimens cured at low temperature (2 °C together with reduced pressure of 0.1 and 0.01 bar) was not possible. Nevertheless, the resulting properties for very low pressure even at high temperature are not sufficient.

Another component property that strongly affects the mechanical properties is the ratio between silicon and aluminum in the mixture. Mechanical strength, porosity, and elastic modulus all depend on this ratio. If the Si/Al ratio is low (below 1.4), the microstructure of the resulting cement is reported to be highly porous and inhomogeneous, resulting in inferior mechanical properties. The ratio above 1.9 shifts the particle balance in another direction, lowering the mechanical properties as the ratio increases further. By staying within the ratio range from 1.65 to 1.9, the porosity is low and the material bulk is largely homogeneous, which translates into high mechanical strength and elastic modulus. Such behavior can be explained by the chemical balance between the particles, where skewed particle ratios result in a high number of unreacted particles, creating defects inside the final product.

In this paper, the optimal silicon to aluminum ratio seemed to be achieved at 45% binder concentration mixed with regolith at a 70% regolith-to-binder ratio. Modifying the binder mixture by, for example, adding a hydroxide, might also shift the balance, for better or for worse. It is important to track this ratio in order to optimize the mixture properties and to ensure the best possible performance.

In the current binder composition iteration, a large amount of binder with respect to regolith is required to incorporate all of the powder into a mixture. The lowest ratio of binder to regolith that was achieved was 30/70, and the resulting mixture was very viscous and inconsistent at times. Such a ratio is very high when it comes to transportation costs of the binder from Earth to Mars. While potentially required elements can be extracted from various sources on Mars and processed into binder components, technologically, such an idea is far from being possible. From this point of view, it is required to modify the binder composition in such a way that a considerably lesser quantity of it is necessary.

When it comes to printing, a large number of phenomena come into the picture that complicate the process. It was found out that, under pressure during deposition, the mixture becomes compressed and that it rapidly reacts, which makes it unprintable. This was expected for such a reaction, as oligomers are forced to reallocate into the most compact structure possible, reacting with each other. This effect caused the mixture to be deposited unevenly, skewing the results and complicating the printing process. Delays between the layers deposited by the printer also affect the properties in a negative way, as the material bulk becomes inhomogeneous. The viscosity of the mixture is highly dependent on the regolith content, and even a difference of 2% makes a significant difference when it comes to printing. The limited outlet diameter of the syringe that could be achieved during the project (7 mm) also limited the ratios of regolith to binder that could be explored to obtain data. As expected, changing the manufacturing method from molding to 3D printing introduces multiple issues that have to be solved during mission design.

### 5.2. Geopolymer Binder Improvement

When it comes to the degree of regolith powder processing, smaller particles allow for faster overall dissolution of aluminosilicate particles in aqueous medium. The powder used in this research was preprocessed into fine particles before packaging and shipping, meaning the dissolution during the mixture creation was fast and homogeneous. However, in real application conditions, regolith powder on Mars might contain very large particles together with small ones, increasing not only the dissolution speed, but also introducing a further discrepancy in the reaction speed between the various regions in the material bulk. This might lead to non-homogeneous final properties, which is not good for overall mechanical properties. This is easy to bypass, however, as a grinder machine can be used in order to reduce the overall particle size, potentially together with a sieving machine.

Optimally, the binder should include a strong base as one of the components, for example, sodium hydroxide NaOH. It serves multiple purposes in the geopolymer mixture, namely introducing easily accessible hydroxide particles OH^−^, which can be incorporated in the monomers, increasing alkalinity of the mixture, thereby speeding up the reaction, and also acting as a catalyst during the gelation phase. However, for the purpose of this research, it was decided to omit the usage of hydroxide. First of all, working with a strong base is more dangerous, and considering experiments with 3D printing where the pressure involved is required, mishandling the solution could potentially have a negative influence on the printer and its surrounding. In addition, since it was planned to make a large number of specimens at the same time, making a solution with one soluble material was faster and more convenient. Adding a base into the picture would also introduce another variable into the specimen matrix, yielding less useful information in the end due to time restrictions. Overall, using a simpler solution provides better baseline results for determining the dependency on the main variables of the reaction: temperature and pressure.

Expanding on the previous point, higher alkalinity does not always directly improve the resulting mixture properties [39]. In fact, high alkalinity induces the formation of ${\left[SiO_{2}{\left(OH\right)}_{2}\right]}^{2-}$. The former particles tend to form smaller oligomers with $2Al{\left(OH\right)}_{4}^{-}$ and halt further polymerization, while the latter allow for more links to be formed, and hence increase the size of the resulting polymer. This means that the balance between ${\left[SiO_{2}{\left(OH\right)}_{2}\right]}^{2-}$ and ${\left[SiO{\left(OH\right)}_{3}\right]}^{-}$ particles directly impacts the mechanical properties of the formed geopolymer cement, which in turn directly correlates with the alkalinity of the solution. This is one of the reasons why a combination of a base and a silicate is typically used over a pure hydroxide.

### 5.3. Application to Mars

In the current iteration, the mechanical properties of the mixture after being cured in either low pressure or low temperature are unsatisfactory, and it is safe to assume that decreasing the temperature and pressure even further to simulate Martian conditions would make the situation even worse. Low pressure has to be counterbalanced by significantly increasing the reaction rate, which is achieved by either increasing the temperature or modification of the mixture. Low temperature has to be counterbalanced by helping the particles to arrange into efficient packing, which is achieved by either increasing pressure or, again, modification of the mixture.

When looking at potential solutions in the context of Martian application, increasing the temperature can be achieved by choosing a landing location where the highest temperatures are achieved (around 20 °C). This temperature is achieved in equatorial regions during the Martian midday in summer. However, the curing process takes well over 24 h, and day on Mars lasts 24 h 37 min, meaning the lowest temperatures, as expected during the night, will also be present during the course of curing. Hence, an additional heating mechanism is required. This can be provided in the form of lamps radiating heat or by creating an enclosed dome, which can also provide pressure increase. The former option consumes a large amount of energy, but otherwise requires little equipment to be used. The latter option requires a dome that is larger than the planned building, which requires a very large quantity of resources to be brought/made. A heating lamp or even a sintering laser can be viable solutions to this problem. The dome, assuming it would be used for printing purposes only, seems like too large of an investment to be justified. Overall, in order to reduce the quantity of resources during 3D printing on Mars, the binder mixture has to be modified by at least adding a hydroxide into the mixture, as suggested in the literature. It must be noted that experimental tests and/or numerical simulations must also be carried out to investigate the high-speed impact resistance of the selected printing materials to small space objects. Explicit finite element approaches have shown promising results for high-speed impacts [40,41,42,43,44].

When it comes to printer design, a crucial parameter that has to be maximized is the outlet nozzle diameter. A large diameter allows for thicker paste, which reduces the amount of binder required and decreases the flow, allows for lower deposition pressure, which is needed to ensure the paste remains uncured, and allows for layers of higher thickness to be deposited, which increases mechanical properties of the complete structure. It is worth noting that the delays between layer deposition are inevitable because the structure to be printed has the size of a building, making the printing process considerably longer than when printing small laboratory specimens. The printer for space missions is expected to be designed from scratch, so incorporating a nozzle diameter that is as large as possible should not create issues.

## 6. Conclusions

In this paper, a geopolymer mixture with the implementation of Martian regolith was investigated as a potential construction material for a Martian base. The goal of this research was to study the effect of reduced temperature and pressure on the physical and mechanical properties of cured geopolymer material. The other objective was to look at the properties of the geopolymer mixture with respect to 3D printing, as this is the manufacturing method that looks very promising when it comes to space construction without human interference. The results of testing the specimens cured at room conditions and at higher temperature were very satisfactory, with good robustness and high average flexural strength (>2.5 MPa). The proposed mixture has a chance to be a viable space building material solution with enhanced composition or processing conditions. The results of this study showed that the properties of the mixture can be improved by either increasing temperature, hence speeding up the reaction, or increasing pressure, thus slowing down the boiling out of water. Another improvement to the mixture can be increasing the alkalinity of the solution, which significantly speeds up the reaction, and is typically achieved by adding a strong hydroxide to the binder solution. The basis laid in this research can lead to an effective way of utilizing Martian regolith for unmanned 3D-printing purposes with minimal investment.

## Figures and Tables

**Figure 1 materials-16-05175-f001:**
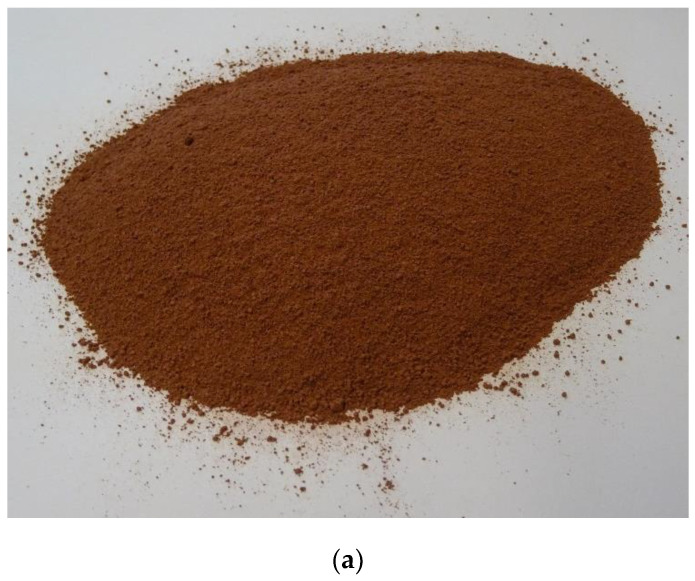
(**a**) Mars simulant regolith powder. SEM images of regolith powder with various particle sizes: (**b**) large (>0.125 mm) sieved particles (×100), (**c**) medium (0.063 mm < x < 0.125 mm) sieved particles (×100), (**d**) small (<0.063 mm) sieved particles (×200), and (**e**) non-sieved particles (×200).

**Figure 2 materials-16-05175-f002:**
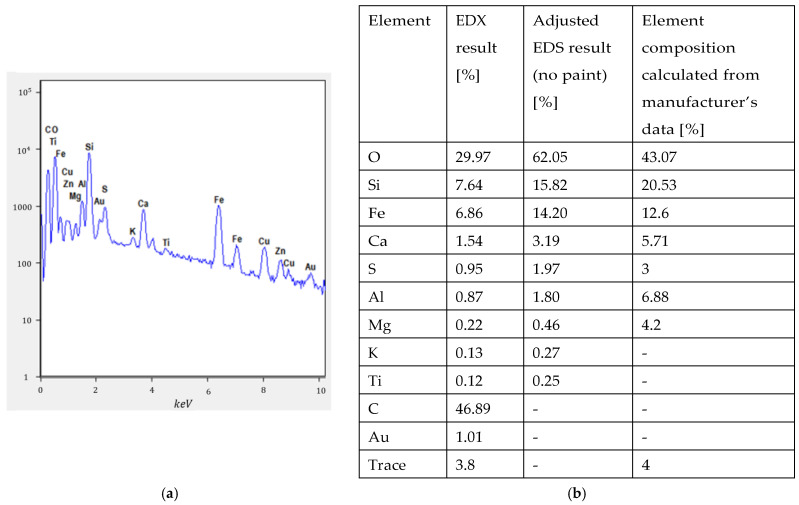
(**a**) EDX results from an MMS-2 regolith powder sample, and (**b**) the tabulated EDX result.

**Figure 3 materials-16-05175-f003:**
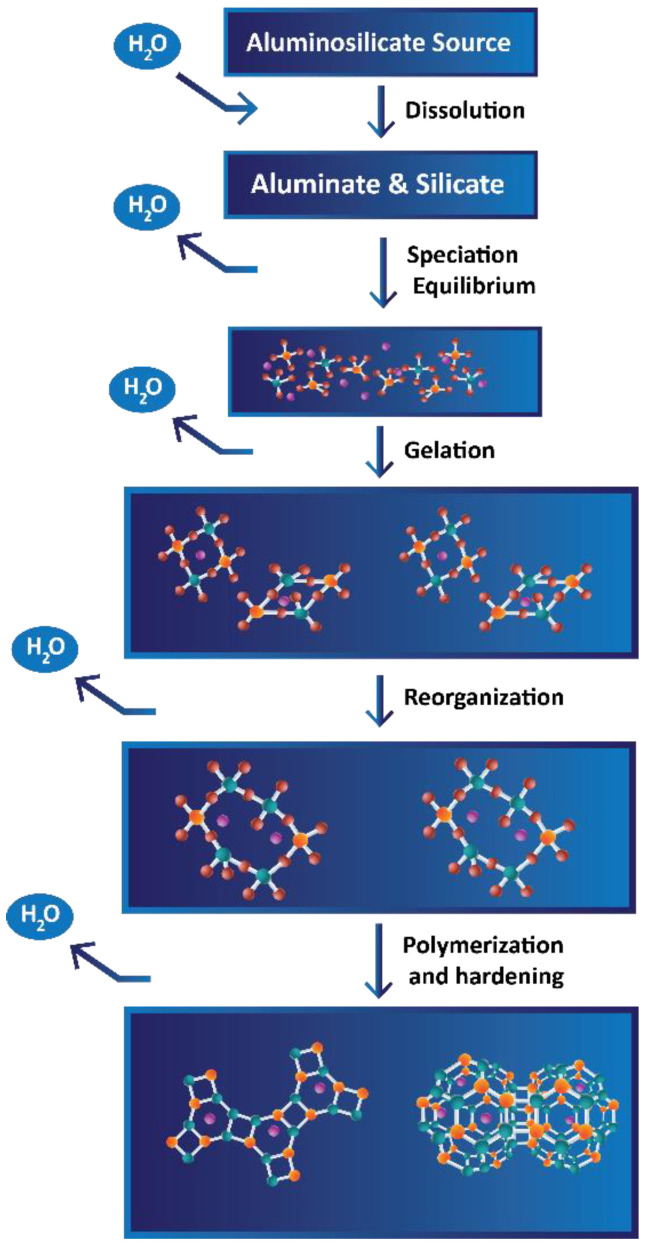
Schematic representation of geopolymerization reaction.

**Figure 4 materials-16-05175-f004:**
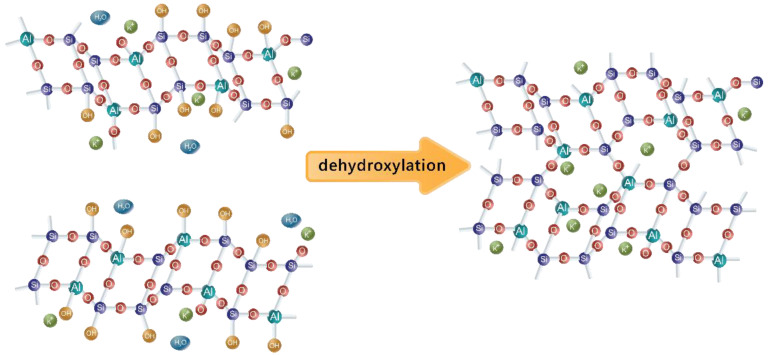
The last step of forming a 3D polymeric network within geopolymer reaction (with potassium instead of sodium) [34].

**Figure 5 materials-16-05175-f005:**
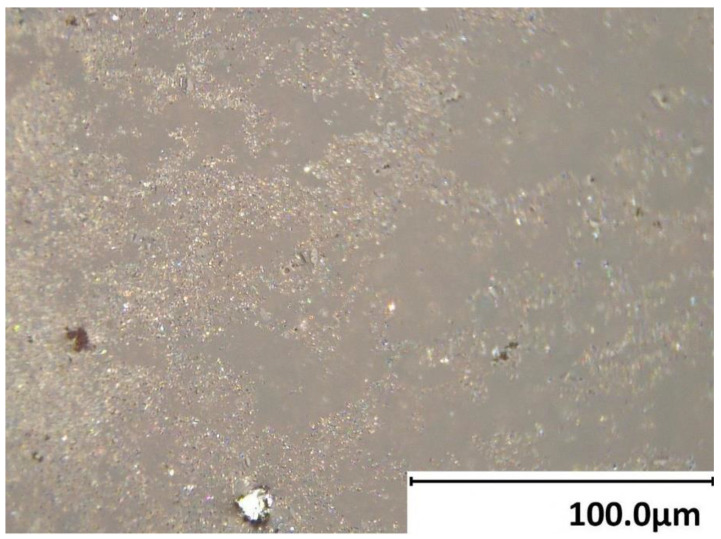
Microscopic image of a mixture surface during saturation and gelation phases (×1000).

**Figure 6 materials-16-05175-f006:**
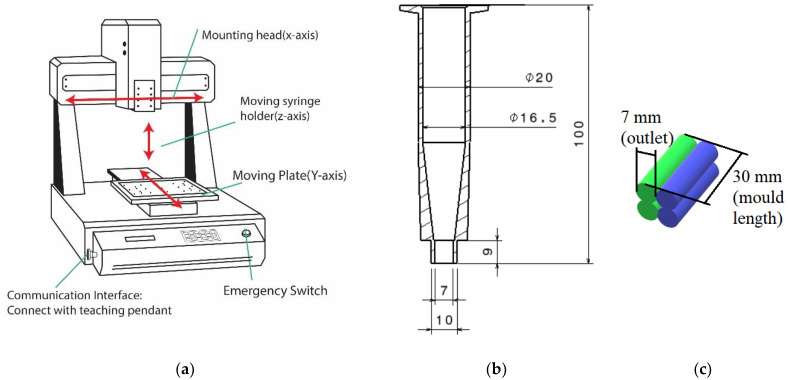
(**a**) Schematic representation of selected printer model with key components indicated [37], (**b**) redesigned extruder syringe, and (**c**) printing pattern for the 3D-printed specimens. Green lines are printed towards the front, while blue lines are printed towards the back. The top two levels are either deposited immediately or after 30 min delay, to test the mechanical property dependence on deposition delays.

**Figure 7 materials-16-05175-f007:**
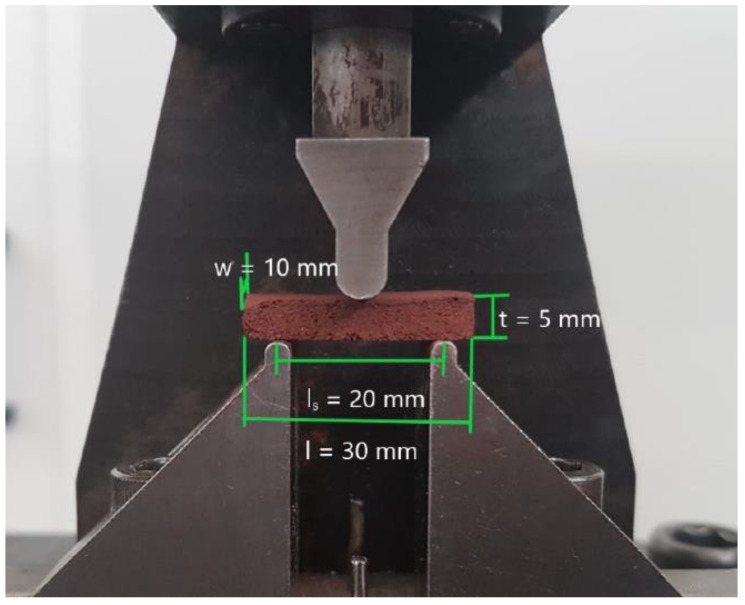
Three-point bending test setup.

**Figure 8 materials-16-05175-f008:**
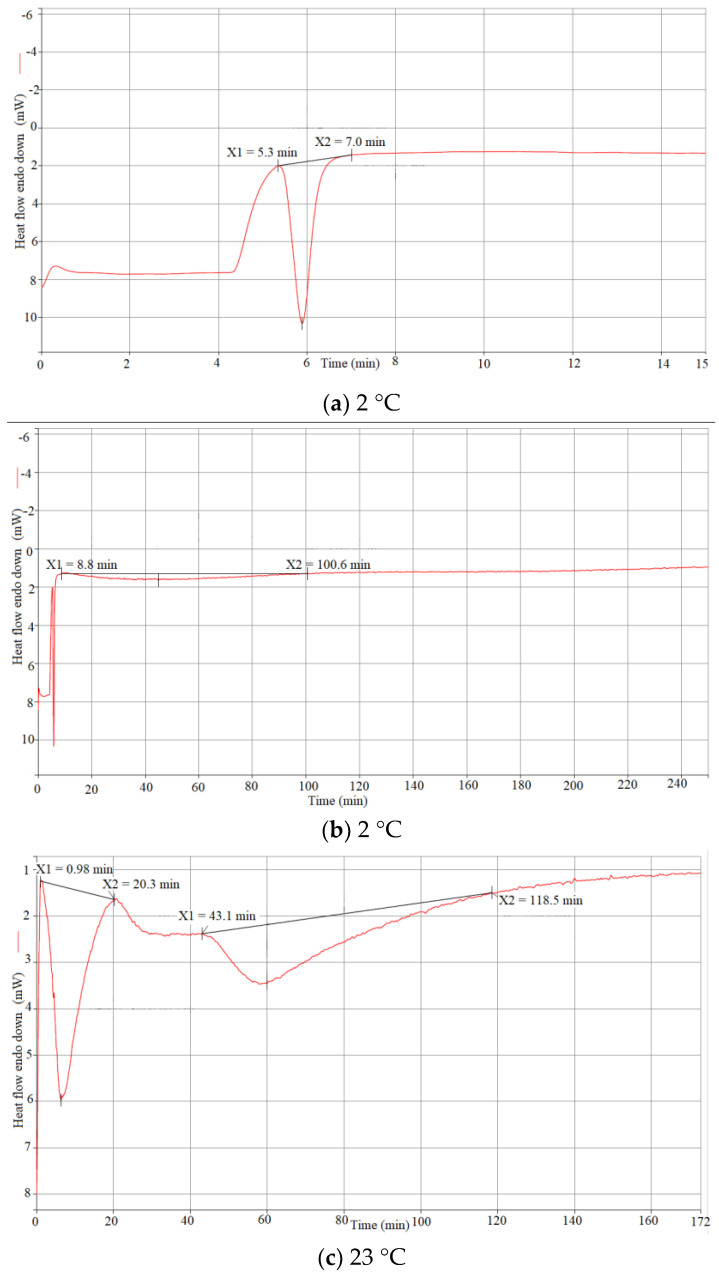
Complete DSC plot for curing the specimen at (**a**,**b**) 2 °C, (**c**) 23 °C, and (**d**) 60 °C, including analysis of both curing steps.

**Figure 9 materials-16-05175-f009:**
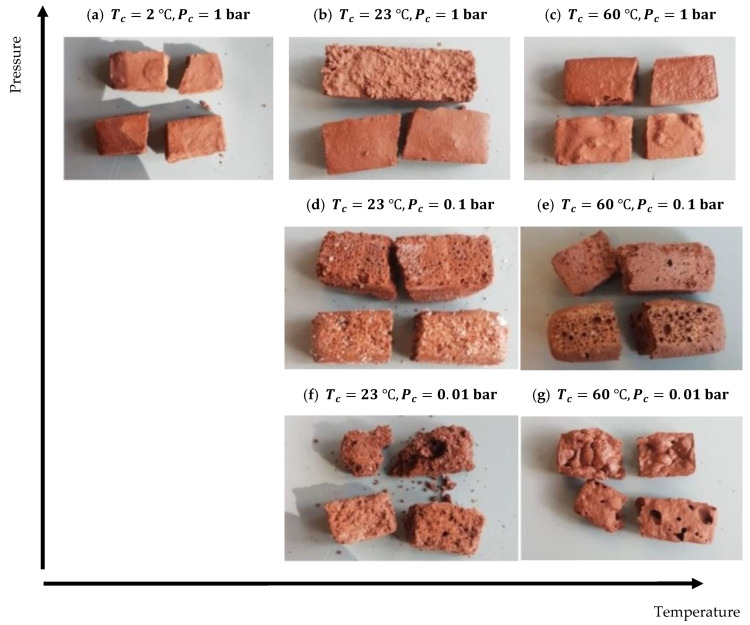
Specimens manufactured under different environmental conditions.

**Figure 10 materials-16-05175-f010:**
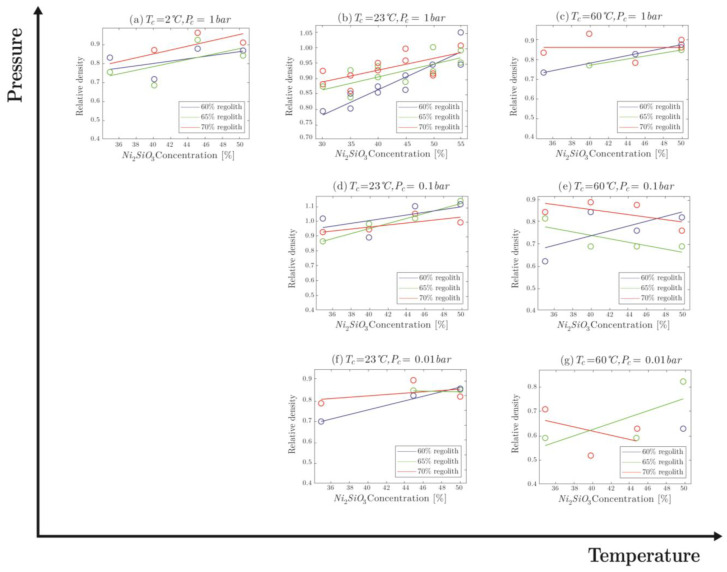
Comparison of relative density values for molded specimens.

**Figure 11 materials-16-05175-f011:**
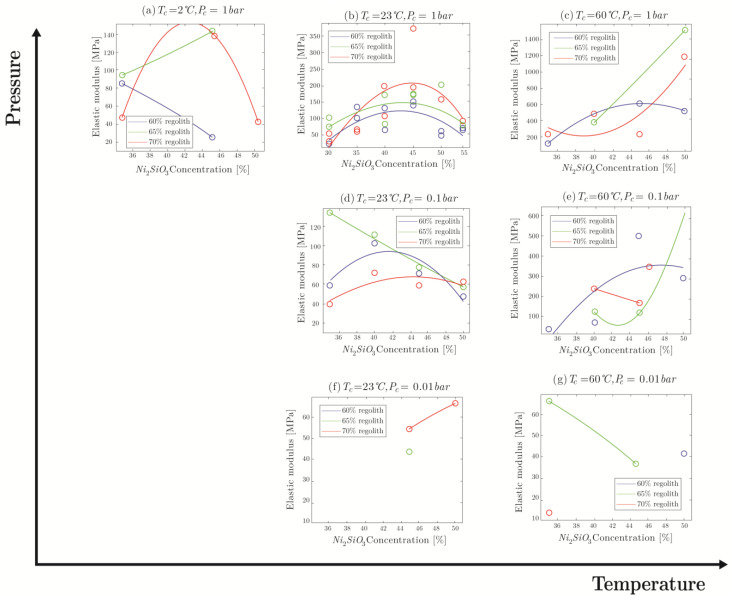
Comparison of ultimate flexural strength values for molded specimens.

**Figure 12 materials-16-05175-f012:**
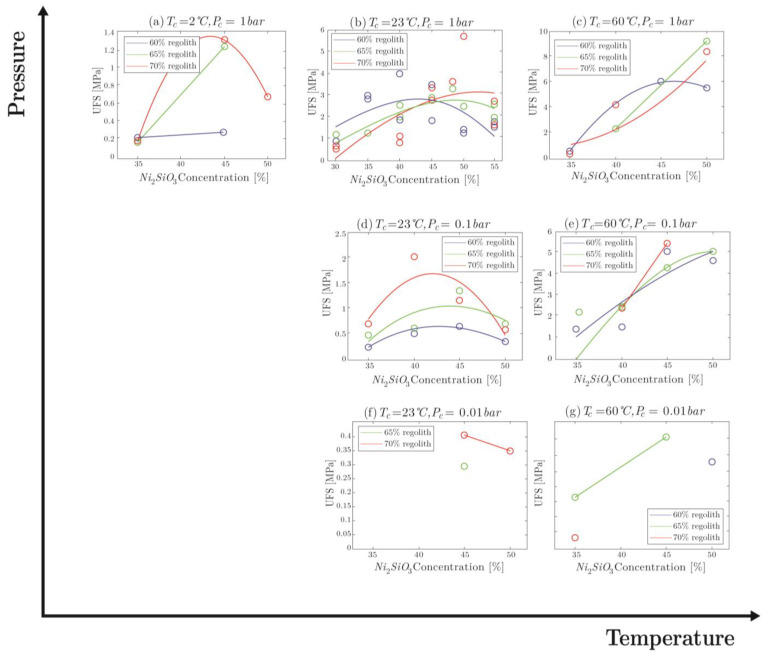
Comparison of ultimate flexural strength values for molded specimens.

**Figure 13 materials-16-05175-f013:**
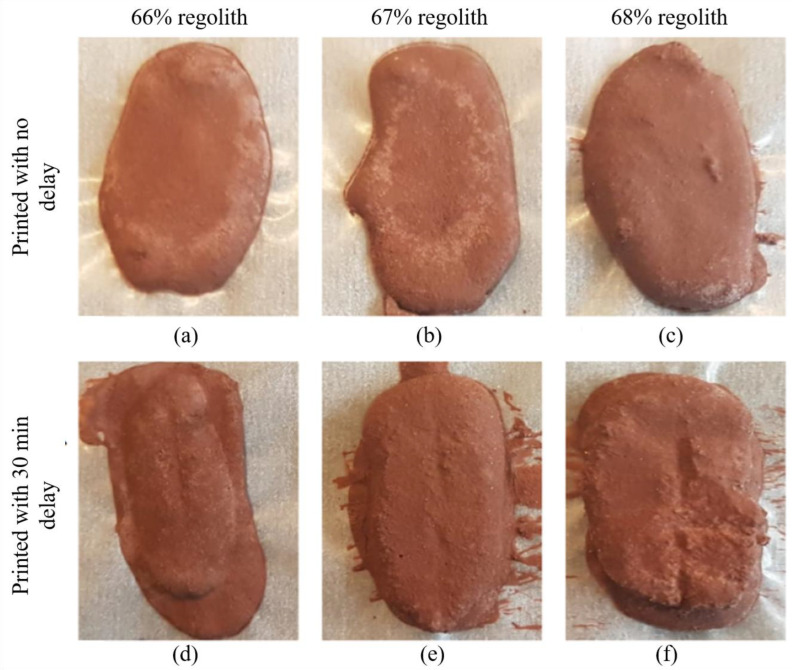
3D-printed specimens with different compositions and with/without delay. (**a**) printed with no delay (66% regolith); (**b**) printed with no delay (67% regolith); (**c**) printed with no delay (68% regolith); (**d**) printed with 30 min delay (66% regolith); (**e**) printed with 30 min delay (67% regolith); (**f**) printed with 30 min delay (68% regolith).

**Figure 14 materials-16-05175-f014:**
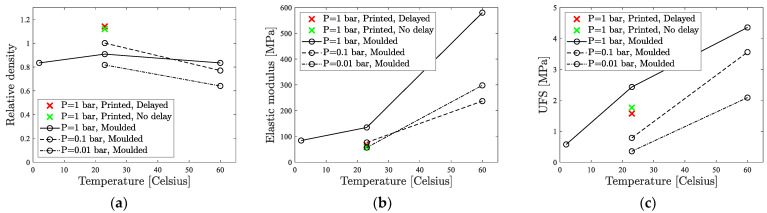
Average (**a**) relative density, (**b**) elastic modulus, and (**c**) ultimate flexural strength (UFS) for all the specimens at different temperature and pressure conditions.

**Table 1 materials-16-05175-t001:** Space 3D-printing challenges.

Challenge	Description
**Material availability**	The costs related to transporting payload are currently very high, and the time required to set up and perform a delivery mission can be also high depending upon destination.
**Meteoroid strikes**	Low amount of atmosphere on Mars does not offer enough protection against meteoroids. Meteorite impact has a potential to strike a built habitat or equipment, damaging it and impairing functionality.
**Temperature variance and near-vacuum conditions**	High temperature variance inflicts high thermal stresses, which cause the material to expand and shrink, resulting in high fatigue loads. Low pressure and high temperature cause the binder to evaporate easier, while low temperature does not allow for binder to react.
**Low gravity**	Combination of low gravity and low pressure can interfere with the binder deposition, especially if the distance between a regolith layer and printer nozzle is large.
**Autonomy**	The distance to Mars is large, meaning signal traveling time is 8.6–24 min, thus obstructing continuous communication. Hence, a trade-off has to be made between full autonomy with no humans present, and partial autonomy with either humans present or slow Mars–Earth communication.

**Table 2 materials-16-05175-t002:** The mass percentage of particles of different sizes from dry sieving tests on a bulk of regolith powder.

Sieve Size [µm]	>125	63–125	<63	Total
**Mass [g]**	17.38	38.86	26.52	82.76
**Mass [%]**	21	47	32	100

**Table 3 materials-16-05175-t003:** Specimen composition matrix for curing in room conditions.

	Na_2_SiO_3_ Concentration [%]	30	35	40	45	50	55
Regolith Concentration [%]							
60		Two specimens per each condition manufactured (36 specimens in total)					
65							
70							

**Table 4 materials-16-05175-t004:** Specimen composition matrix for conditions other than room conditions.

	Na_2_SiO_3_ Concentration [%]	35	40	45	50
Regolith Concentration [%]					
60		One specimen for each combination (12 specimens in total)			
65					
70					

**Table 5 materials-16-05175-t005:** Viability of tested specimens with respect to cure conditions. Green cells indicate good conditions, yellow indicate conditions that produce somewhat useful material, and red indicate poor cure conditions.

	*T_c_* [°C]	2	23	60
*P_c_* [bar]				
1		◦ very low mechanical properties ◦ low robustness ◦ no porosity ◦ long cure time	◦ good mechanical properties ◦ good robustness (most compositions) ◦ no porosity ◦ medium cure time	◦ good mechanical properties ◦ decent robustness (most compositions) ◦ no porosity ◦ fast cure time
0.1		N/A	◦ poor mechanical properties ◦ good robustness ◦ some porosity ◦ medium cure time	◦ decent mechanical properties ◦ decent robustness (most compositions) ◦ high porosity ◦ fast cure time
0.01		N/A	◦ poor mechanical properties ◦ unacceptable robustness ◦ high porosity ◦ medium cure time	◦ decent mechanical properties ◦ low robustness ◦ high porosity ◦ fast curing time

## Data Availability

The data presented in this study are available on request from the corresponding author.

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
