# Peer review of "3D Printing of Habitats on Mars: Effects of Low Temperature and Pressure"

_materials, 2023, doi:10.3390/ma16145175_

Round 1

Reviewer 1 Report

The 3D printing in a simulated Mars environment is very interesting for future in-situ resource utilization. In this paper, the effects of low temperature and pressure are studied, and interesting results are presented. However, there are still many drawbacks. My detailed comments are as follows.

1.         The simulated Mars environment have many aspects, including temperature, pressure, and low gravity, and so on. The authors should clearly state why they only choose to study the effects of temperature and pressure.

2.         Although there is few reports in the field of 3D printing on Mars, some reviews related to the additive manufacturing of norganic non-metallic materials are suggested to cite in Introduction part:

Chemical Engineering Journal, 2023, 463: 142378.

Journal of Advanced Ceramics, 2021, 10637-674.

3.         Actually, the 3D printing difference between on Earth and Mars, the authors are suggested to list the advs or disadvs, or differences in a table.

4.         Regarding the tables and figures in this manuscript, the authors are suggested to re-draw and give the tables and figures with more professional quality for scientific publication.

5.         The testing standards for the strength test are suggested to provide.

should be improved.

Author Response

We appreciate the positive feedback from the reviewer and thank you for providing valuable comments that increase the quality of our manuscript. Responses to the comments are attached.

Reviewer 2 Report

The manuscript "3D printing habitats on Mars: effects of low temperature and pressure" submitted for publication on Materials has been reviewed. It deals with an experimental proposal for manufacturing components by 3D printing utilizing a readily available material such as regolith. 

The manuscript is novel and interesting, well organized. Engilsh is fine.

In my opinion I suggest the following major revisions:

1) Abstract length is excessive in comparison with the journal requirements (max 200 words). Please check.

2) line 77, [15-24]. 10 references, 1 shot. Unacceptable. Please split and motivate each of them in the text.

3) Fig. 1) - c in the caption is presentdd twice. Please check.

4) Fig. 2 please add captions (a), (b). P Check teh composition (the sum is not 100%).

5) Fig. 3 taken from [31]: did the authors get the right for reproduction?

6) Fig. 4 taken from [32]: did the authors get the right for reproduction?

7) Line 317: 3-D printing process. Please check

8) Fig. 6 taken from [35]: did the authors get the right for reproduction?

9) Fig. 9 Please enlarge pictures, increase the quality and the focus.

10) Fig. 10 f) and g) It is not clear why 2 curves instead of 3 are reported in the graphs.

11) In Fig. 11 same considerations of Fig. 10.

12) In Fig. 12 same considerations of Fig. 10.

13) Conclusions must be improved.

After that the manuscript can be reconsidered for publication on Materials.

Author Response

(The authors gave the same response as above.)

Round 2

Reviewer 2 Report

The manuscript has been improved and can be accepted in the present form.